# Ultrafast enzymatic digestion of proteins by microdroplet mass spectrometry

Xiaoqin Zhong[1], Hao Chen[2] & Richard N. Zare [1]✉

Enzymatic digestion for protein sequencing usually requires much time, and does not always result in high sequence coverage. Here we report the use of aqueous microdroplets to accelerate enzymatic reactions and, in particular, to improve protein sequencing. When a room temperature aqueous solution containing 10 µM myoglobin and 5 µg mL$^{-1}$ trypsin is electrosonically sprayed (−3 kV) from a homemade setup to produce tiny (~9 µm) microdroplets, we obtain 100% sequence coverage in less than 1 ms of digestion time, in sharp contrast to 60% coverage achieved by incubating the same solution at 37 °C for 14 h followed by analysis with a commercial electrospray ionization source that produces larger (~60 µm) droplets. We also confirm the sequence of the therapeutic antibody trastuzumab (~148 kDa), with a sequence coverage of 100% for light chains and 85% for heavy chains, demonstrating the practical utility of microdroplets in drug development.

[1] Department of Chemistry, Fudan University, Shanghai 200438, China. [2] Department of Chemistry and Environmental Science, New Jersey Institute of Technology, Newark, NJ 07102, USA. ✉email: rnz@fudan.edu.cn

Reaction-rate acceleration in micron-sized droplets (microdroplets) was first observed by mass spectrometrists during the processes of desorption electrospray ionization (DESI)[1] and also by electrospray ionization (ESI)[2,3]. Microdroplets can be generated by a number of setups including microfluidics[4], surface drop-casting[5], theta-tip capillaries[6,7], paper spray[8,9], or other different spray-based ionization methods[10,11]. Striking reaction accelerations with these methods were reported. In the past few years, microdroplets have been extensively reported to dramatically accelerate various types of single-phase or two-phase organic reactions that have slow kinetics or need specific catalysts in the bulk phase[12–14]. They have also been used to study fast reaction kinetics via microdroplet fusion mass spectrometry (MS)[6,10,15,16], to perform chemical syntheses[17], and also to facilitate nanomaterial synthesis[11,18,19]. The exact reasons for the reaction rate acceleration in microdroplets have not been clearly established, but it is commonly believed that it is caused mainly by the sharp difference between the environments of microdroplets and the corresponding bulk phase[20]. Various factors may contribute, such as droplet size, surface charge[21], reagent confinement[22], solvent composition, and droplet evaporation. In addition to the achievement in organic synthesis, microdroplets are of interest in promoting biochemical reactions because of the gentleness of the process and particularly because aqueous microdroplets provide a benign environment that is compatible with life[23]. However, the application of microdroplets to biochemical analysis has seldom been investigated. We report here several examples demonstrating how the tryptic digestion of proteins can be accelerated in water microdroplets.

To identify proteins by the bottom-up proteomics strategy, enzymatic digestion of proteins is an essential and critical approach to break down proteins into smaller polypeptides prior to protein sequence elucidation by mass spectrometry[24,25]. During a typical procedure, the protein solution is mixed with a proper amount of some enzyme, such as trypsin, and incubated overnight at 37 °C[26]. To facilitate digestion, protein denaturation is usually performed before the digestion to destroy the compact, globular structure and expose more proteolytic cleavage sites[27]. Commonly used methods include the application of external stress or additives, such as heat, radiation, or urea[28,29]. In addition, reductive alkylation is often used to eliminate disulfide bonds[28]. To further accelerate protein digestion, various attempts have been taken to reduce the incubation time from overnight to several minutes, including increasing the digestion temperature[30,31], using columns or porous materials for trypsin immobilization[32–35], addition of organic solvents[36,37], applying microwave energy[38] or focused ultrasonic field[39], or some combination of these[40]. Here we present an alternative approach involving no pre-treatment of the sample. We use room-temperature microdroplet chemistry to achieve simple and nearly complete protein digestion in less than 1 ms (herein digestion refers to one or more cleavages at amino acid residues that are capable of scission by the enzyme of choice).

In the present work, we demonstrate that microdroplets generated during electrosonic spray ionization (ESSI)[41,42] and directly coupled with a mass spectrometer (microdroplet-MS) can achieve complete online cleavage of a relatively large peptide when the travel distance is increased to 2 cm. As an example, we investigate the tryptic digestion of the protease-resistant protein myoglobin, which has 153 amino acid residues and a heme group with iron in its center. The results demonstrate the advance and potential significance of microdroplets in protein identification, including a dramatic decrease in digestion time—from overnight with traditional methods or several minutes with other accelerated devices—to less than a millisecond for the complete cleavage of peptide bonds at the C-terminal side of lysine or arginine residues except when followed by proline; and an increase in sequence coverage from 60 to 100%. The possible mechanism driving this acceleration is discussed, including the surface concentration effect in microdroplets, the spontaneous generation of hydroxyl radicals or hydrogen peroxide, gas bubbles induced by ammonium bicarbonate, and the chain ejection model (CEM) during the ESI process[43] to drive the proteins to the microdroplet surface. To further demonstrate the practical utility of this technology, the sequence of the therapeutic antibody trastuzumab (~148 kDa), also known as Herceptin, is confirmed by ESSI-MS with a sequence coverage of 100% for light chains and 85% for heavy chains. The higher sequence coverage for heavy chains compared to that obtained from bulk-phase digestion (74%) suggests the superiority of microdroplets in improving protein enzymatic digestion, especially for large or protease-resistant proteins.

## Results

**Performance optimization of microdroplet-MS.** Figure 1 presents a schematic diagram of the experimental apparatus for microdroplet-MS. The microdroplets containing the protein and enzyme dissolved in an ammonium bicarbonate buffer were generated by electrosonic spray in which a sheath of rapidly flowing dry $N_2$ gas surrounds a capillary held at typically ±3 kV. To sequence the peptide of interest, collision-induced dissociation (CID) was applied for the fragmentation of the isolated precursor ion with an isolation width of 1 mass to charge ratio ($m/z$) and optimized collision energy of 25 arbitrary manufacturer's units under full scan mode. The protein sequence was acquired from the UniprotKB database with its specific accession numbers. Peptides resulting from digestion were identified by comparison of observed peptide molecular weights with theoretical values using MS-digest program from Protein Prospector version 5.19.1 (University of California, San Francisco, CA, USA) to perform an in silico digest of the protein of interest. A more complete description of how the data is recorded is presented in the Methods section.

Adrenocorticotropic hormone (ACTH, accession number: Q01718) is a polypeptide consisting of 39 amino acids. It is produced by the front of the pituitary gland in the brain and its function is to regulate levels of the steroid hormone cortisol released from the adrenal gland. We employed the tryptic digestion of ACTH as a simple model system for optimizing the performance of microdroplet-MS. A stream of microdroplets was produced by our homemade sprayer using pressurized nebulizing $N_2$ gas at 120 psi for the nebulization of a 5-mM solution of ammonium bicarbonate containing 10-μM human ACTH (1-24) and 5 μg mL$^{-1}$ trypsin. The initial droplet size generated by ESSI was reported to be around 6 μm in diameter[10], and also tested by a laser particle analyzer to be around 9 μm in diameter (see

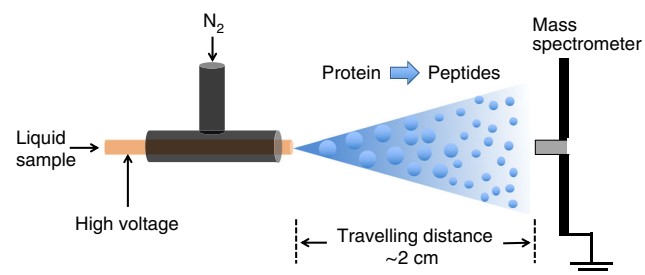

**Fig. 1 Schematic of the experimental apparatus for the online proteolysis by microdroplet chemistry coupled with mass spectrometry (ESSI-MS).** The inner capillary has an i.d. of 50 μm and an o.d. of 148 μm to which a high voltage (typically +3 kV or −3 kV) is applied.

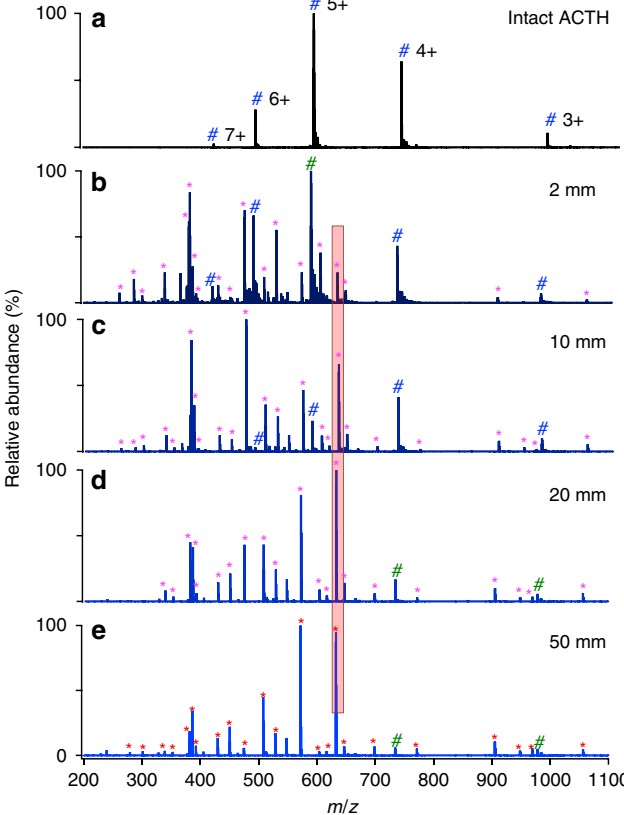

**Fig. 2 Mass spectra of 10-μM human ACTH (1–24) in 5-mM aqueous NH₄HCO₃ sprayed by the homemade sprayer. a** undigested (no trypsin); **b–e** digested with 5 μg mL⁻¹ trypsin at different travel distances between the sprayer tip and MS inlet for 2, 10, 20, and 50 mm, respectively. Magenta asterisks denote the peptide fragments, and blue #s mark undigested ACTH peaks.

droplet size measurement results in Supplementary Method 1 and Fig. 1). The microdroplets traveled in the air at a speed of 84 ± 18 meter per second characterized by a high-speed optical camera[10]. During the flight of microdroplets into mass spectrometer, the protein digestion was accelerated significantly. Once inside the heated inlet, the microdroplet evaporates and the reaction stops[10], which is verified by changing the temperature of the heated inlet and observing no obvious change in the extent of digestion. However, the reaction extent in microdroplets is found to vary significantly by changing the traveling distances of microdroplets, which shows what happened inside the MS inlet has no or little effect on the enzymatic digestion reaction in our case. Therefore, the digestion time in the microdroplets could be determined by varying the traveling distance between the sprayer tip and the MS inlet. As shown in Fig. 2, digestion progressed as the travel time to the mass spectrometer inlet increased.

The digestion extent was found to correlate directly to the microdroplet travel distance, which can be seen by examining Fig. 2b–e, where the digestion yield of ACTH was obviously improved by increasing the traveling distance from 2 mm to 20 mm. Twenty-six peptide fragment peaks were successfully identified, fully covering the whole sequence of ACTH, as listed in Supplementary Table 1. When the distance was increased to 50 mm, corresponding to a travel time of 0.6 ms based on the previously reported microdroplet velocity of 80 meter per second[10], ACTH ion peaks appearing in Fig. 2e are tiny, indicating most of ACTH has been cleaved one or more times. With other experimental factors fixed, the dependence of

digestion yield of a chosen peptide on microdroplet travel distance was roughly evaluated by comparing as a function of travel distance the peak intensity ratio of the most abundant peptide in Fig. 2 at $m/z$ 632.4 to the intensity sum of all the peaks from the intact multicharged ACTH, including $(ACTH + 3H)^{3+}$, $(ACTH + 4H)^{4+}$, $(ACTH + 5H)^{5+}$, $(ACTH + 6H)^{6+}$ and $(ACTH + 7H)^{7+}$, as shown in Supplementary Fig. 2. The relative intensity of intact ACTH decreased greatly from 2 mm to 20 mm and then changed slightly even at a much longer distance of 50 mm, which means most ACTH has been cleaved at a distance of 20 mm. Although the extent of digestion was greater at 50 mm than 20 mm, the highest peak intensity at 50 mm became much weaker (4.8E6 for Fig. 2e), while it did not change too much (from 2.6E7 in Fig. 2b to 1.6E7 in Fig. 2d) when increasing the distance from 2 mm to 20 mm. Consequently, 20 mm was selected as the optimal travel distance for both efficient digestion and good detection. The sequence of this peptide was confirmed by CID tandem MS (MS/MS) analysis (as exemplified in Supplementary Fig. 3).

As a control, ACTH was also digested with trypsin and the digestion products were analyzed using standard electrospray ionization mass spectrometry (ESI-MS, LTQ Orbitrap Elite, Thermo Scientific fitted with a commercial heated ESI probe). Owing to the bigger size of the commercial ESI sprayer tip (500 μm in diameter) and the lower sheath gas flow rate (around 10 psi) compared to the N₂ pressure of 120 psi used for ESSI, the initial droplets generated by the commercial ESI source were measured by a laser particle analyzer to be around 60 μm in diameter, as shown in SI-1. The initial microdroplet size plays a very important role in digestion acceleration rather than the droplet shrinkage or fission during ESI or ESSI. In a review by Kebarle and Tang[44] on how long it takes for enough evaporation to occur to achieve the first Coulomb fission of a typical droplet, they estimated the time to be about 450 ms for a pure methanol droplet, which is longer than the droplet flight time in a typical ESI or ESSI-MS experiment. In a study on heptane droplets by Gomez and Tang[45], they experimentally find that heptane droplets with a diameter of 4.7 μm and a charge density of 113.1 C m⁻³, corresponding to nearly a million charges, it takes about 527 μs for the droplet to evaporate to the point where the first droplet fission is expected to occur. For aqueous droplets produced by commercial ESI source in our case, the time required for the first droplet fission to occur is, of course, even longer due to the lower vapor pressure of water than either methanol or heptane. During the ESI process with a commercial source, microdroplet acceleration of digestion is very small due to the limited flight time of droplet with such an initial large size. Thus only a slight amount of digestion was observed in Fig. 3a, which is consistent with the behavior reported for the spontaneous generation of hydrogen peroxide in aqueous microdroplets[46]. Because the commercial ESI-MS causes negligible digestion acceleration, it was employed for the analysis of peptides from the digestion of the same solution in bulk solution at 37 °C. After 3 h, the bulk-phase digestion gave a nearly comparable result (Fig. 3b) to that obtained by microdroplet-MS (Fig. 2d), and only 16 peptide peaks were detected, as listed in Supplementary Table 2.

Figures 2 and 3, taken together, demonstrate the feasibility of microdroplet-MS for ultrafast digestion of peptides. Other experiments were performed to demonstrate that the temperature of the mass spectrometer inlet and the voltage value applied to the spray capillary had little effect on the digestion process, although higher charge states in mass spectra imply that proteins are more unfolded and denatured, and thus possibly to digest more easily. However, a high MS inlet temperature (>200 °C) with a proper spray voltage value (>2.5 kV with our setup) is helpful for the

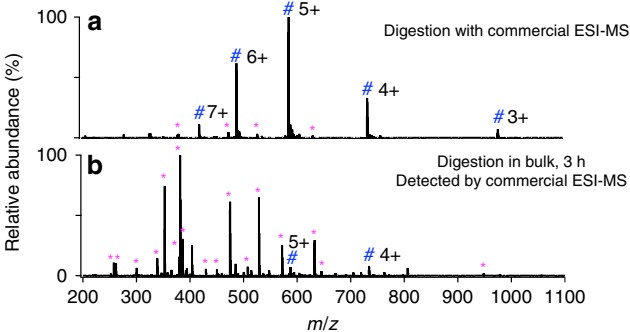

**Fig. 3 Comparison of ACTH digestion with various methods. a** commercial ESI-MS (LTQ Orbitrap Elite, Thermo Scientific), **b** bulk phase at 37 °C for 3 h, followed by analysis with commercial ESI-MS. The 16 peptide peaks found in **b** are listed in Supplementary Table 2. Magenta asterisks denote the peptide fragments, and blue #s mark undigested ACTH peaks. The unassigned peaks could be peptide adduct ions with solvent or background solvent peaks.

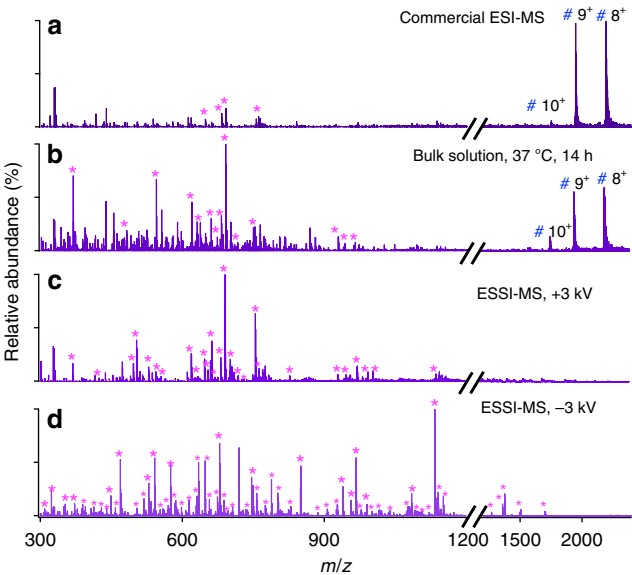

**Fig. 4 Mass spectra of myoglobin digestion with various methods.**
**a** Commercial ESI-MS (LTQ Orbitrap Elite, Thermo Scientific), **b** bulk phase at 37 °C for 14 h, followed by analysis with commercial ESI-MS, and ESSI-MS applied with **c** a positive high voltage (+3 kV), and **d** a negative high voltage (−3 kV). Magenta asterisks denote the peptide fragments, and blue #s mark undigested myoglobin peaks.

detection of peptides in the digests. A negative potential was found superior to a positive potential, probably due to the more compatible pH value (slightly basic) for protein digestion. Even when no potential is applied, digestion still occurs, but to a lesser extent.

**Protease-resistant protein digestion.** Having demonstrated that microdroplet-MS is able to markedly accelerate digestion, it is natural to inquire whether it could achieve digestion of proteins that have proved particularly recalcitrant to tryptic digestion. Myoglobin (accession number: P68082) is such a protein having 153 amino acid residues. Figure 4a presents the peptide fragments from myoglobin seen in standard ESI-MS and Fig. 4b those observed by standard ESI-MS after 14 h of digestion in bulk solution at 37 °C. These should be compared to the results from

the microdroplet-MS technique (Fig. 4c), where we found that the time for myoglobin digestion could be dramatically reduced from overnight to less than 1 ms. From Fig. 4c and Supplementary Table 3, 31 peaks corresponding to 19 peptides were identified, and a higher sequence coverage of around 86% was obtained compared to that of around 60% from only 13 identified peptides with a typical procedure as described in the Methods section. To find the lost sequences, a negative high voltage (−3 kV) was applied for ESSI ionization instead of the positive high voltage owing to the more compatible pH value with the tryptic digestion. Negative ion acquisition mode was set for the MS analysis. Surprisingly, 55 peaks corresponding to 38 peptides were identified. This corresponds to 100% sequence coverage. By matching the experimental results with the results of an in silico digest, all theoretical-cleavable peptide bonds after K and R except when following by proline due to the steric hindrance were found to be broken, as shown in Fig. 4d and Supplementary Table 4. Moreover, trypsin digested nearly all of the myoglobin in the initial starting solution under microdroplet-MS conditions. From the results, we found, besides the more compatible pH environment for the protein digestion, ammonium bicarbonate also provides better detection under negative ion mode. The mass spectra we report are mostly from singly negatively charged peptides which produce less background interference, making the mass spectrum less complicated and more easily interpretable.

As a powerful tool for protein digestion, microdroplets were further applied for the digestion of cytochrome *c* (accession number: P62895), with sequence coverage around 83% under a positive voltage of 3 kV shown in Supplementary Fig. 4 and Supplementary Table 5 and 100% under a negative voltage (−3 kV) shown in Supplementary Fig. 5 and Supplementary Table 6.

**Post-PAGE gel protein digestion.** To demonstrate further the practicality of this technique in proteomics, two proteins, cytochrome *c* and α-casein (accession number: P02662 and P02663, in a mixture were separated first by 15.5% SDS polyacrylamide gel electrophoresis (PAGE)[47], as shown in Fig. 5c. Then, protein bands stained with Coomassie blue dye were excised from the gel and subjected to digestion by microdroplet MS after a treatment procedure as described in the Methods section. The mass spectra gave sequence coverage of around 90.3 and 99% for αS1-casein and cytochrome *c*., respectively, as shown in Fig. 5a, b, and Supplementary Table 7.

**Synthetic peptide digestion with a slow kinetic constant.** The most ideal case in enzymatic digestion for proteomics study is achieved when all cleavage sites are digested, but in practice, enzymes often fail to cleave all scissile bonds, even though the reaction time is sufficiently long. This failure to achieve complete coverage is mainly attributed to neighboring amino acids around the cleavage sites. The presence of acidic residues, glutamate (E) or aspartate (D), near the cleavage site was reported to reduce the proteolysis speed significantly by forming salt bridges with the basic arginine (R) and lysine (K) and inhibiting the approach of R or K to the complementary aspartic acid at the bottom of the trypsin active site[48]. In our case, microdroplets could completely cleave the synthetic peptide sample (LYAA-[DTR]-LYAVR, 10-μM in 5 mM NH4HCO3) reported with a very low kinetic constant $(0.24 \times 10^{-3}\,s^{-1})$[49], shown in Fig. 6, which is mainly attributed to the remarkable acceleration of proteolysis speed by microdroplets despite the negative influence of acidic microdroplets on trypsin digestion. From this result, we can expect microdroplet-MS to be a powerful proteolysis tool that can easily cleave most theoretically scissile bond and produce less missed cleavage peptides. However, when the acidic residue of D is

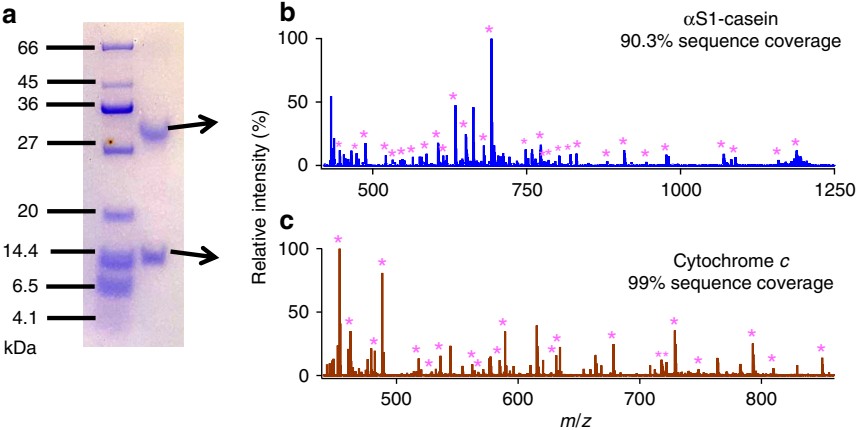

**Fig. 5 Trypsin microdroplet digestion of proteins with ESSI-MS (+3 kV) after PAGE gel separation. a** PAGE gel showing the protein ladder and two protein bands stained with Coomassie blue. Mass spectra showing the microdroplet digestion of **b** α-casein and **c** cytochrome c. Magenta asterisks denote the peptide fragments.

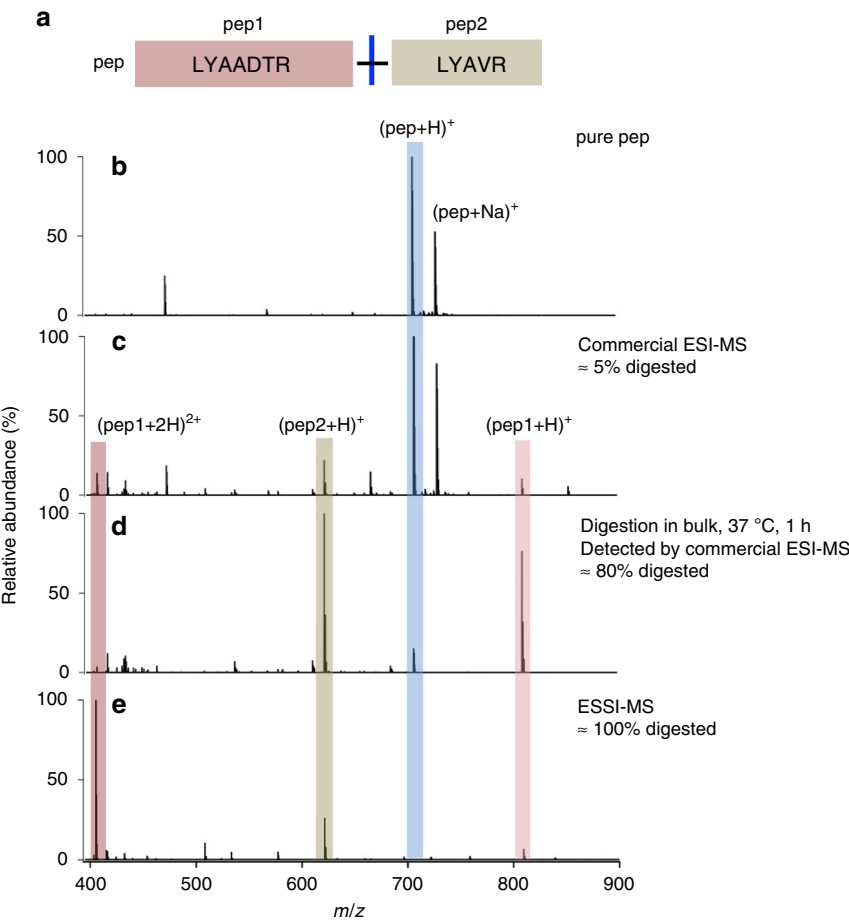

**Fig. 6 Trypsin digestion of a synthetic peptide with various methods. a** Peptide sequence, **b** mass spectrum of pure peptide. Mass spectra of the peptide digested with **c** commercial ESI-MS, **d** in bulk phase at 37 °C for 1 h, followed by analysis with commercial ESI-MS, and **e** ESSI-MS applied with a high voltage (+3 kV).

directly followed by K, the synthetic peptide (LYAA-[DK]-LYAVR, 10 μM in 5 mM aqueous $NH_4HCO_3$) failed to be digested by microdroplets (data not shown) due to the closer position of D to K. Besides, K is indirectly attached to the active site of trypsin through a water molecule and the lower pH value in microdroplets may inhibit the formation of a water-molecule bridge between K and the active site of trypsin.

The pH range that allows the microdroplet-MS to maintain the excellent acceleration effect for the proteolysis was also studied in Supplementary Fig. 6. We found when the pH was lowered to less than 4, the accelerated digestion by microdroplet-MS was inhibited, with only 40% of the synthetic peptide being digested. When the pH was elevated to 11, the microdroplet-MS could still digest the synthetic peptide completely.

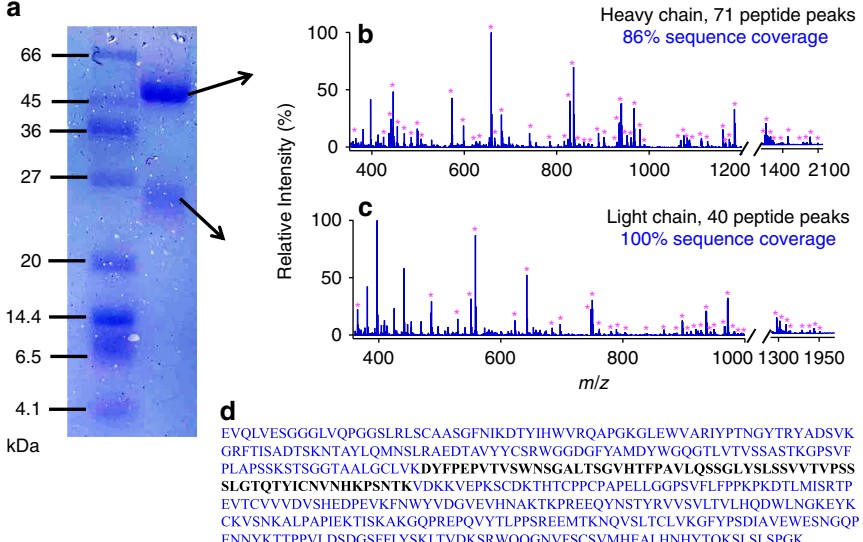

**Fig. 7 Trypsin microdroplet digestion of the therapeutic antibody trastuzumab with ESSI-MS (−3 kV) after PAGE gel separation. a** PAGE gel showing the protein ladder, antibody's light and heavy chain bands stained with Coomassie blue. Mass spectra showing the microdroplet digestion of **b** light chain and **c** heavy chain. **d** The heavy chain of trastuzumab, with the covered sequence marked in blue and the uncovered sequence in black. Magenta asterisks denote the peptide fragments.

**Therapeutic antibody sequence confirmation.** Sequence confirmation of a therapeutic antibody was demonstrated herein to show the potential utility of our method in antibody-based drugs' validation, because therapeutic antibodies have revolutionized the treatment of many types of diseases, such as cancer and auto-immune disorders. Trastuzumab (accession number: P04626) is a humanized monoclonal antibody that has been clinically used to treat patients with invasive breast cancers. After being treated with a procedure described in the Methods section, trastuzumab and trypsin were mixed and immediately sprayed for digestion and detection using ESSI-MS. Satisfactory antibody sequence coverage was achieved with cleavage of 100% for the light chain and around 86% for the heavy chain, as shown in Fig. 7a, b and Supplementary Table 8. In our case, the trypsin-cleavable sites limited the sequence coverage of trastuzumab's heavy chain, and there is only one K residue in the long sequence not found by ESSI-MS, marked in black in Fig. 7c. We found that its sequence coverage could be further improved to 98% by using another unspecific enzyme, elastase (Supplementary Fig. 7 and Supplementary Table 9). We found ESSI-MS microdroplet digestion could generate peptides with longer length and improve the sequence coverage of heavy chains from 74% in conventional bulk digestion overnight under 37 °C to 86% as shown in Fig. 7, Supplementary Fig. 8, and Supplementary Table 10. Due to its fast speed and efficient digestion, our technique offers a major advantage over traditional methods of peptide mapping in the identification of the protein sequence, and it could potentially meet the increasing demand for monoclonal antibody sequence confirmation to support downstream drug development in the biopharmaceutical industry.

## Discussion

In protein sequencing by the bottom-up proteomics strategy, protein digestion is a lengthy step and remains one of the bottlenecks in terms of time consideration. Table 1 lists various approaches for accelerated protein digestion. As shown in Table 1, microdroplets when coupled with MS gave the least digestion reaction time, if not including the sample preparation time. Digestion in microdroplets consumes tiny samples ($0.16\ \mu L\ s^{-1}$) and could be easily extended to other setups producing small microdroplets, such as nano-

**Table 1 Comparisons of various techniques for accelerated protein digestion[68].**

| Accelerated technique | Online | Digestion time |
|---|---|---|
| High temperature | possible | ~15 min |
| Microwave | possible | ≤15 min |
| Ultrasound | Not feasible | ≤5 min |
| High pressure | Yes | <1 min |
| Infrared | Not done | ~5 min |
| Organic solvent | Not done | ≤5 h |
| On-column immobilized enzyme | Yes | <6 min |
| On-chip immobilized enzyme | Yes | 5 s |
| Magnetic particle immobilized enzyme | Yes | ~30 s |
| Microdroplets (this work) | Yes | <1 ms |

Note: Digestion time refers to digestion reaction time, not the total experiment time.

electrospray ionization (nanoESI) sprayer. Indeed, we have tested nanoESI and it also works well for protein digestion. Nevertheless, it needs to be emphasized that protein digestion is just one step in a many-step process involving protein sequencing, and often it is not the sole rate-limiting step especially in terms of analyses of complicated biological samples. Because our setup also acts as a MS emitter, we believe that it provides a convenient interface to couple with other sample preparation steps. However, its practical utility will need to be judged on how well it can be integrated into the other steps, and this assessment is likely to be valid only case by case. Herein, as one of the most promising potential applications, we demonstrated the advantages of our technology in the sequence confirmation of a branded therapeutic antibody (trastuzumab), including fast speed, easy use, and high sequence coverage, as shown in Fig. 7.

It has already been proposed that reactions could be catalyzed by the heterogeneous environment between vapor and condensed phases or between different solvents in contact[50–52], although experimental investigations are seldom carried out. The pH of a water microdroplet differs from bulk, and there remains no consensus whether the surface of the microdroplet is acidic or basic, although the preponderance of evidence supports the idea that $OH^{-}$ preferentially goes to the surface. The pH may even

vary gradually among the surface and the interior of a micro-droplet[53–55]. The Zare research group has suggested that water molecules at and near the air-water interface autoionize more readily into $H^+$ and $OH^-$ than in bulk water, caused by the lack of three-dimensional hydrogen bonding and the presence of a strong electric field. It is known that a higher $H^+$ concentration could facilitate acid hydrolysis of proteins and a higher $OH^-$ concentration would promote trypsin activity. The value of the surface potential at the air-water interface is still debatable and estimated to be on the order of tens of megavolts per cen-timeter[56]. A potential of approximately 3 V across the 5 Å air–water interface[57,58] exceeds a standard potential of 1.23 V for the electrolysis of water molecules. It also exceeds the potential of 2.72 V for removing electrons from hydroxide ions to form hydroxyl radicals in bulk phase[57]. The pKa at the air-water interface has been reported by Francisco and co-workers who found that the redox potential is different from the bulk one[52,59–61]. Colussi and co-workers have also reported chemical reactions at the vapor-water interface, water-hydrophobic inter-face, and air-water interface with distinctly different properties than the same reactions in bulk solution[62–64].

In fact, the microdroplet periphery provides a more energeti-cally favorable environment for redox reactions, and we thus hypothesized that the $OH-$ at the air-microdroplet interface might more readily release its electron and be oxidized to form hydroxyl radicals. We have supported this postulation by several tests using salicylic acid to capture hydroxyl radicals[65]. We also found and assayed the spontaneous generation of $H_2O_2$ in microdroplets (1–20 μm in diameter) by the $H_2O_2$-sensitive fluorescent dye peroxyfluor-1[46]. In the present case, to test whether the formation of reactive oxygen-containing species, such as hydroxyl radicals (·OH) and hydrogen peroxide mole-cules, at the air-water interface may promote protein digestion, we added hydrogen peroxide (30 μM) to a solution of myoglobin for bulk digestion. We found that the digestion time could be reduced to 4.5 h with 90% sequence coverage, as shown in Sup-plementary Fig. 9 and Supplementary Table 11.

As an ESI-MS compatible buffer to improve solution pH sta-bilization, Konermann and coworkers proposed that ammonium bicarbonate could facilitate protein unfolding during the final ESI stages by outgassing $CO_2$ and forming bubbles, which are well known to denature proteins[66]. We compare the digestion effi-ciency of two proteins, cytochrome $c$ and myoglobin, in different buffers, as shown in Supplementary Fig. 10. The Supplementary Fig. 10a, b show us that good digestion of cytochrome $c$ could be obtained either in ammonium bicarbonate or ammonium acetate, and their digestion extent was comparable. While for the diges-tion of myoglobin with a rigid structure, ammonium bicarbonate gave a better digestion efficiency than ammonium acetate, as shown in Supplementary Fig. 10c, d, which might be consistent with the reference reported by Konermann and coworkers[66] that ammonium bicarbonate could facilitate protein unfolding during the ESI by outgassing $CO_2$ and forming bubbles, which are well known to denature proteins.

They also proposed a chain ejection mode (CEM) to account for the protein ESI behavior[67] and afterwards found evidence from atomistic simulation and ion mobility spectrometry in 2018[43]. Through CEM, unfolded proteins are driven to the dro-plet surface by hydrophobic and electrostatic factors, which may act synergistically with surface accumulation effects in micro-droplets[20] to improve protein digestion.

Clearly, more work is needed to establish in detail the factors that account for this markedly enhanced digestion power of water microdroplets, but the outline of how special microdroplets are compared to bulk solution is clearly emerging. This is a topic of ongoing research.

## Methods

**Microdroplet-MS with ESSI**. A stream of microdroplets was generated by infusing an aqueous sample solution containing peptide or protein (10 μM) and trypsin (5 μg mL$^{-1}$) in 5 mM ammonia bicarbonate ($NH_4HCO_3$, pH 8) or 5 mM ammonia acetate ($NH_4OAc$, pH 8) with a syringe at a flow rate of 10 μL min$^{-1}$ into a homemade sprayer. Ammonia bicarbonate, ammonia acetate and all the protein reagents were obtained from Sigma-Aldrich (Shanghai, China). Deionized water (18.2 MΩ cm) was prepared by the Milli Q purification system (Millipore Advantage A10) and used in all aqueous solutions.

The sample solution was sprayed from the tip of a fused silica capillary (148 μm o.d., 50 μm i.d., Polymicro Technologies, China) of the homemade sprayer and assisted by a nebulizing gas of dry $N_2$ with a pressure of 120 psi. By placing the sprayer in front of a high-resolution mass spectrometer (LTQ Orbitrap Elite, Thermo Scientific, San Jose, CA) at a proper position, the microdroplets were directed into MS for real-time analysis when applying a positive or negative high voltage (±3 kV, BOHER HV, Genvolt, U.K.) to the sprayer. The MS inlet capillary was always maintained at 275 °C and capillary voltage at 0 V. No other source gases were used when digestion was performed in microdroplets.

For control tests, protein or peptide was also digested using a traditional procedure. 10 μM adrenocorticotropic hormone from human (ACTH, 1–24, Genscript, China) or 100 μg mL$^{-1}$ proteins were first denatured by heating at 95 °C for 5 min and then were incubated with 5 μg mL$^{-1}$ of trypsin in a 5 mM $NH_4HCO_3$ buffer, pH 8, under 37 °C. Aliquots of 100 μL were taken at different reaction times for freezing at −20 °C to stop the reaction and were further submitted to standard ESI-MS analysis.

To sequence the peptide of interest, MS/MS with collision-induced dissociation (CID) was applied for the fragmentation of the isolated precursor ion with an isolation width of 1 $m/z$ and optimized collision energy of 25 arbitrary manufacturer's units under full scan mode. All the MS1 and MS2 were performed under a resolution of 12000. Data analysis and conversion into exact mass list were performed by Xcalibur Qual Browser (ThermoFisher Scientific, San Jose, CA). The mass spectra were plotted by IGOR Pro (Version 6.00 for Macintosh, WaveMetrics, Lake Oswego, OR, USA).

MS-digest program from Protein Prospector version 5.19.1 (University of California, San Francisco, CA, USA) was applied for an in silico digest of the protein of interest. The search parameters were set as following: database from UniprotKB, trypsin digestion, three maximum missed cleavages, variable modification of oxidation for cytochrome c, myoglobin, and α-casein, variable modifications of oxidation and carbamidomethyl (C) for trastuzumab antibody, and 5 ppm as mass tolerance.

**Commercial ESI-MS**. For the analysis with standard ESI-MS, the samples were also directly infused with a syringe at the flow rate of 10 μL min$^{-1}$ and sprayed from a commercial heated ESI probe with a needle of around 500 μm in inner diameter fitted for a high-resolution mass spectrometer (LTQ Orbitrap Elite, Thermo Scientific, San Jose, CA). The spray was assisted with a sheath gas flow of 10 arbitrary units (10 psi). The temperature of the MS inlet capillary was set at 275 °C and the ESI voltage was set as ±3 kV.

**Droplet size characterization**. The size distributions of microdroplets generated from the homebuilt ESSI sprayer and the commercial heated ESI probe under the conditions described above respectively, were characterized by a laser particle analyzer of HELOS (Hi208, Sympatec GmbH, Suzhou, China). It is worth men-tioning that the aqueous solution containing 5 mM $NH_4HCO_3$ (pH 8) was all infused with a syringe at a flow rate of 20 μL min$^{-1}$ for the droplet size mea-surement, instead of 10 μL min$^{-1}$, because the laser particle analyzer cannot cap-ture enough droplets when a low flow rate of 10 μL min$^{-1}$. The droplet size generated by ESSI sprayer and the commercial ESI probe could be estimated respectively from the graphs by plotting the volume distribution of droplets versus the droplet size, shown in Supplementary Fig. 1.

**Post-PAGE gel protein digestion**. For protein separation by gel electrophoresis, 10 μL of sample solution containing cytochrome c (1 mg mL$^{-1}$) and α-casein (1 mg mL$^{-1}$) were loaded onto 15.5% SDS-PAGE gels. All the setups and reagents for gel electrophoresis were purchased from Sangon (Shanghai, China). Electrophor-esis was carried out at 200 V for 1 h at room temperature. A low range protein ladder was used as the size marker. The gel was stained by Coomassie blue and then destained by a solution containing 30% ethanol and 12.5% acetic acid in $H_2O$. The stained protein bands were excised from the gels and ground into tiny pieces for efficient protein extraction with a commercial kit (Sangon, Shanghai). The sample was sonicated in an ultrasonic water bath for 30 min until the gel pieces turned opaque. The extracted proteins were further purified by performing pre-cipitation in pure acetone (99.9%, Adamas, China) at −20 °C for 3 times and then desalting by a centrifugal filter (Amicon Ultra-0.5, Millipore, USA) with a nominal molecular weight limit of 10 kDa. The purified antibody samples were diluted in 100 μL 5 mM $NH_4HCO_3$ buffer, pH 8 and further submitted to digestion by microdroplet MS. The whole procedure, including protein extraction, purification, and microdroplet digestion, could be normally finished in less than 1 h.

**Trastuzumab sequencing**. For the separation of antibody's light and heavy chains by gel electrophoresis, 10 μL of sample solution containing trastuzumab (1 mg mL$^{-1}$, MedChemExpress, USA) and 1X commercial protein loading buffer (Sangon, Shanghai, China) was loaded onto 12% SDS-PAGE gels. Electrophoresis was carried out at 200 V for 1 h at room temperature. A conventional range protein ladder was used as the size marker. The light and heavy chains of trastuzumab were extracted from their stained bands respectively on gel with the commercial kit (Sangon, Shanghai, China). This kit consists of two kinds of buffer, one is an aqueous solution containing surfactant for protein extraction from PAGE gel, another is an organic solution for protein precipitation. Detailed components were not provided by the vendor. Then, iodoacetamide (≥99%, Sigma, China) was added into the extraction buffer to a final concentration of 100 mM and incubated for another 30 min at room temperature in dark to block the thiol groups in the antibody chains. Finally, sample solutions containing light and heavy chains were purified respectively through a similar procedure described in the above paragraph and the purified chains were diluted in 100-μL 5 mM NH$_4$HCO$_3$ buffer, pH 8. The digestion process for antibody light and heavy chains by microdroplet MS or in the bulk phase is same as those for other proteins demonstrated above. Besides the specific digestion with trypsin, the enzyme elastase was similarly applied for trastuzumab unspecific digestion aiming at finding the uncovered sequence. The sample solution containing the extracted antibody chains and elastase (5 μg mL$^{-1}$) in 5 mM NH$_4$HCO$_3$, pH 8 was infused with a syringe at a flow rate of 10 μL min$^{-1}$ into the microdroplet MS setup for digestion and identification.

**Reporting summary**. Further information on research design is available in the Nature Research Reporting Summary linked to this article.

## Data availability

There are no restrictions for the raw data associated with the figures presented and they are available from the corresponding authors on request. We have given accession codes for the proteins we have studied.

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

## Acknowledgements

This work was supported by the Scientific Research Startup Foundation (IDH1615113) of Fudan University, NSF (CHE-1915878), and key postdoc funding (KLH1615199).

## Author contributions

X.Z., H.C., and R.N.Z. designed the research. X.Z. performed research, analyzed the data, and wrote the first draft of the paper. All authors discussed the results and edited the manuscript.

## Competing interests

The authors declare no competing interests.
