## [Peer Review File · Nature Communications]

Reviewers' comments:

Reviewer #1 (Remarks to the Author):

'Ultrafast enzymatic digestion of proteins by microdroplet mass spectrometry' with authors Zhong, Chem, and Zare is a highly interesting and potentially useful method for fast protein digestion. The manuscript is well written and will be of interest to the journal's readers. However, there are a couple of aspects that are bothersome to this reviewer. One is the emphasis on obtaining digestion in 1 ms. If all of the digestion occurs in the droplets, then the time of digestion is close to 1 ms, but the time to do the digestion, which is what the method is compared against, is longer. The authors should note this in the direct comparison. Another issue which seems more difficult to rationalize is why this does not work with a conventional ESI source. Most measurements of ESI droplet size in the literature report droplets at least as small as the 9 μm droplets the authors produce with ESSI. Certainly the 60 μm droplets for ESI reported in this paper is an outlier number. Further, Cooks has a paper in the October issue of JASMS (vol 30 p 2022-2030 on "Reaction Acceleration in Electrospray Droplets: Size, Distance, and Surfactant Effects" which show that reaction acceleration occurs with ESI droplets. This paper should be cited and the authors should discuss why their approach would or would not work with ESI or nanoESI. A final point is that even if larger droplets are produced in the author's setup, they must decrease in size to produce the bare ions observed, so why don't they accelerate the digestion reaction during this desolvation process?

Line 340 needs a period after surface.

Line 435 needs more description of the commercial protein extraction kit.

Reviewer #2 (Remarks to the Author):

This paper reports the results of protein digestion and the sequencing of the therapeutic antibody trastuzumab (~148 kDa), with a sequence coverage of 100% for light chains and 85% for heavy chains conducted in microdroplets. This offers yet more support for the application of microdroplet chemistry, this time accelerating the enzymatic reaction in microdroplets and also improving the protein sequencing protocol compared to standard methods.

Thus the work is highly suitable for publication in Nature Communication.

However, there are a few points that I suggest for revision

(1) On line 126, it says "Once inside the heated inlet, the microdroplet evaporates and the reaction stops." Is there some way to verify this?

(2) Line 186-190, is mentioned the "other experiments". What are they specifically? A reference(s) should be provided if it is not conducted in this work.

(3) Should there be a more general conclusion at the end of the paper?

(4) Line 347-348 "A potential of approximately 3 V across the 5 Å air-water interface..." In the reference, the surface potential calculated are summarized in Table 1. You can see that the largest number is -890mV, which is far from 3V. How would this reference support the statement in the current work?

(5) It is not clear from the manuscript why the authors chose an ammonium bicarbonate buffer solution. In addition, in order to reveal if ammonium bicarbonate contributes to the digestive mechanism as suggested by the authors it would be beneficial to see the effects of a different solvent on the microdroplet enzymatic acceleration.

(6) The authors report that the water droplets shrink during the spraying process which contributes to the reaction acceleration. However, the authors don't provide any evidence to support this claim. It would be useful if the authors can provide some details on how much the droplet shrinks during the process and why they claim that the shrinking process facilitates the reaction acceleration? Is it just collision frequency?

(7) The authors claim that the polarity of the voltage has little effect and that even no voltage was effective but to a less digestive extent. I am wondering if a possible digestive mechanism could be that the high ionic state promotes denaturation of the protein which facilitates the reaction acceleration. In figure 2 it is evident that the proteins with the higher ionic states 5, 6 and 7 are cleaved at a faster rate than the proteins with lower charge state 3 and 4. This seems contradictory.

Reviewer # 1

Recommendation: Ultrafast enzymatic digestion of proteins by microdroplet mass spectrometry' with authors Zhong, Chen, and Zare is a highly interesting and potentially useful method for fast protein digestion. The manuscript is well written and will be of interest to the journal's readers. However, there are a couple of aspects that are bothersome to this reviewer.

Response: We thank the reviewer very much for these positive comments.

Comment 1. One is the emphasis on obtaining digestion in 1ms. If all of the digestion occurs in the droplets, then the time of digestion is close to 1ms, but the time to do the digestion, which is what the method is compared against, is longer. The authors should note this in the direct comparison.

Response:

Yes, we agree with the reviewer that the 1 ms refers only to the digestion time in the microdroplets and that's exactly what we compared in the manuscript. The whole protein identification process in our case also includes the sample solution preparation, infusion into the ESSI sprayer, and then spraying the microdroplets into the mass spectrometer for detection. All these steps normally take a few minutes to do. As the reviewer suggested, we have noted this fact in a footnote for Table 1 that "digestion time" refers to "digestion reaction time", not the total time for performing the experiment in the revised manuscript.

Comment 2. Another issue which seems more difficult to rationalize is why this does not work with a conventional ESI source. Most measurements of ESI droplet size in the literature report droplets at least as small as the 9 μm droplets the authors produce with ESSI. Certainly the 60 μm droplets for ESI reported in this paper is an outlier number. Further, Cooks has a paper in the October issue of

JASMS (vol 30 p 2022-2030 on “Reaction Acceleration in Electrospray Droplets: Size, Distance, and Surfactant Effects” which show, that reaction acceleration occurs with ESI droplets. This paper should be cited and the authors should discuss why their approach would or would not work with ESI or nanoESI. A final point is that even if larger droplets are produced in the author’s setup, they must decrease in size to during this desolvation process?

Response:

Although the reviewer might be surprised by the size of the microdroplets from the commercial source (LTQ Orbitrap Elite, Thermo Scientific, San Jose, CA), we have measured it using a laser particle analyzer (HELOS Hi208, Sympatec GmbH, Suzhou, China), which is the same way we measured the size of the microdroplets from our ESSI source. Please compare the characteristics of the two sources: The commercial source has a 500 μm tip diameter whereas the ESSI source has a 50 μm tip diameter. The commercial source uses a flow rate of 20 $\mu\text{L}/\text{min}$ and the ESSI source uses the same flow rate. This information is stated in the supporting information in SI-1. However, the sheath gas flow rates are not the same. The commercial one is 10 psi whereas the ESSI source is 120 psi. As a consequence, the diameter of the microdroplets from the commercial source has an average value of 60 μm , which is much larger than the average diameter of microdroplets (9 μm) from the ESSI source. As shown in the recently published study on the generation of hydrogen peroxide in aqueous microdroplets (See Fig. 2 of Ref. 47), smaller aqueous microdroplets are much more effective than larger ones. Thus, during the time of flight, the microdroplets from the commercial source cannot be expected to digest as rapidly as the ESSI source. Only a slight amount of digestion was observed in the microdroplets from the commercial source and could be ignored under the conditions we fixed in our case. That’s why we employed it to analyze the peptides from bulk digestion. We have added this information to the main text.

As suggested we have cited the new paper from Cooks and coworkers.

We also tested that nanoESI also works for protein digestion acceleration and we believe that other setups producing small microdroplets will accelerate digestion. We have added this information to the main text.

Comment 3. Line 340 needs a period after surface.

Response: Thanks for telling us about the missing period, which we have added.

Comment 4. Line 435 needs more description of the commercial protein extraction kit.

Response: Description of the commercial protein extraction kit was highlighted in yellow in the revised manuscript as following:

This kit consists of two kinds of buffer, one is an aqueous solution containing surfactant for protein extraction from PAGE gel, another is an organic solution for protein precipitation. Detailed components were not provided by the vendor.

Reviewer # 2

Recommendation: This paper reports the results of protein digestion and the sequencing of the therapeutic antibody trastuzumab (~148 kDa), with a sequence coverage of 100% for light chains and 85% for heavy chains conducted in microdroplets. This offers yet more support for the application of microdroplet chemistry, this time accelerating the enzymatic reaction in microdroplets and also improving the protein sequencing protocol compared to standard methods. Thus, the work is highly suitable for publication in Nature Communication. However, there are a few points that I suggest for revision.

Response: We thank the reviewer very much for these positive comments.

Comment 1. On line 126, it says “Once inside the heated inlet, the microdroplet evaporates and the reaction stops.” Is there some way to verify this?

Response:

Typically, the heated inlet stops the reaction, which can be seen in Figure 3 of Ref 10, which studied the reaction of ascorbic acid with dichlorophenolindophenol.

This behavior was verified by changing the temperature of the heated inlet and observing no obvious change in the extent of digestion. This information is highlighted in yellow in the revised text.

SI-3 also verified that the reaction extent in microdroplets is varied significantly by changing the traveling distances of microdroplets, which shows what happened inside the MS inlet has no or little effect on the enzymatic digestion reaction in our case.

For most enzyme-involved biochemical reactions, temperature is one of the most critical factors to reserve the enzyme activity. In our case, for efficient MS detection, the MS inlet was always set under a very high temperature (275 degree),

whereas we believe it should have inactivated the trypsin activity.

Comment 2. Line 186-190, is mentioned the “other experiments”. What are they specifically? A reference(s) should be provided if it is not conducted in this work.

Response:

We performed some other experiments to test whether the temperature of the MS inlet and the polarity of the voltage applied have big influences on the digestion process. We found both of them had little effect on the digestion process but better detections of the peptides from the digests could only be obtained at a higher MS inlet temperature (> 200 degree) and with a proper voltage value (> 2.5 kV with our setup). Even no potential being applied was also effective, but to a very low detection efficiency and probably lesser digestion extent. However, we found a negative potential was superior to a positive potential for the protein digestion in our case, probably due to the more compatible pH value (slightly basic) to the protein digestion.

For clarity, we changed “other experiments demonstrated...” into “other experiments were performed to demonstrate...”.

Comment 3. Should there be a more general conclusion at the end of the paper?

Response:

Thanks the reviewer for this helpful suggestion and now we have added a “Conclusions” section, highlighted in the revised manuscript as following:

In summary, we have demonstrated that aqueous microdroplets containing trypsin act as a simple, ultrafast, and powerful tool for protein digestion when coupled directly with mass spectrometric detection. The proteolysis-resistant protein, myoglobin, could be fully digested to obtain 100% sequence coverage and 100% cleavage of theoretically cleavable peptide bonds in less than 1 ms, indicating the great advance of the unique environment provided by microdroplets for sample mixing, protein structure alteration and protein backbone cleavage. Digestions of

other samples, including some synthetic peptides that are known to resist tryptic digestion, the proteins isolated from gel electrophoresis, and the therapeutic antibody of trastuzumab, were also presented. The results show the promising utility of microdroplets as a novel, practical, and nearly universal technique for assisting protein sequencing and drug development. This first demonstration of protein digestion acceleration in microdroplets may also be potentially feasible to a wide range of other biological catalyses, and this is a topic of our ongoing research.

Comment 4. Line 347-348 “A potential of approximately 3 V across the 5 Å air-water interface...” In the reference, the surface potential calculated are summarized in Table 1. You can see that the largest number is -890mV, which is far from 3V. How would this reference support the statement in the current work?

Response:

We believe that 3.1 V for the surface potential and 1.5×10^{10} V/m for the maximum electric field is the correct value estimated in our manuscript, based on the erratum (Kathmann et al., JACS, 2009) published for the original paper version (Kathmann et al., JACS, 2008), where the 1.5×10^{10} V/m was the maximum peak value instead of the original value of 8.9×10^7 V/m. To make this clearer to the reader, we cite the original paper and its erratum.

Comment 5. It is not clear from the manuscript why the authors chose an ammonium bicarbonate buffer solution. In addition, in order to reveal if ammonium bicarbonate contributes to the digestive mechanism as suggested by the authors it would be beneficial to see the effects of a different solvent on the microdroplet enzymatic acceleration.

Response:

In our case, ammonium bicarbonate is traditionally used as a MS-compatible buffer due to its easy evaporation, besides the stable and slightly basic pH which provided to fit the optimal digestion condition for trypsin.

SI-15: ESSI-MS (-3 kV) analysis of cytochrome *c* digests in (a) 5 mM ammonium bicarbonate (NH_4HCO_3 , pH 8), (b) 5 mM ammonium acetate (NH_4OAc , pH 8) and myoglobin digests in (c) 5 mM NH_4HCO_3 (pH 8), (d) 5 mM NH_4OAc (pH 8)

As the reviewer suggested, we further tried another popular MS-compatible solvent, ammonium acetate. Its concentration was kept at 5 mM in H_2O and the pH was adjusted to be around 8, as that of 5 mM ammonium bicarbonate (pH 8). The figures in SI-15(a)-(b) show us that good digestion of cytochrome *c* could be obtained either in ammonium bicarbonate or ammonium acetate, and their digestion extent was comparable. While for the digestion of myoglobin with a rigid structure, the ammonium bicarbonate gave a better digestion efficiency than ammonium acetate, as shown in SI-15(c)-(d), which might be consistent with the reference reported by Konermann and coworkers that ammonium bicarbonate could facilitate protein unfolding during the ESI by outgassing CO_2 and forming bubbles, which are well known to denature proteins. Unfolded proteins are driven to the droplet surface by hydrophobic and electrostatic factors, which may act synergistically with the increased collision frequency and surface effects in microdroplets to improve protein digestion.

Such information has been supplemented as SI-15 and highlighted in the revised

supporting information and manuscript.

Comment 6. The authors report that the water droplets shrink during the spraying process which contributes to the reaction acceleration. However, the authors don't provide any evidence to support this claim. It would be useful if the authors can provide some details on how much the droplet shrinks during the process and why they claim that the shrinking process facilitates the reaction acceleration? Is it just collision frequency?

Response:

In a review by Kebarle and Tang (see Ref 45) on how long it takes for enough evaporation to occur to achieve the first coulomb fission of a typical droplet, they estimated the time to be about 450 ms for a pure methanol droplet, which is longer than the droplet flight time in a typical ESI-MS experiment. In a study on heptane droplets by Gomez and Tang (see Ref 46), they experimentally find that heptane droplets with a diameter of 4.7 μm and a charge density of 113.1 C/m^3 , corresponding to nearly a million charges), it takes about 527 μs for the droplet to evaporate to the point where the first droplet fission is expected to occur. For water droplets with its lower vapor pressure than either methanol or heptane, the time required for the first droplet fission to occur is, of course, even longer. Consequently, we have removed this statement that the aqueous microdroplet shrinks from the text and highlighted such information in yellow in the revised manuscript.

In our opinion, the initial droplet size plays a very important role in reaction acceleration rather than the droplet shrink process. The ESSI could generate much smaller initial droplets than commercial ESI tip, in which initial droplets require a longer time to shrink to generate microdroplets as small as that by ESSI. The difference of their initial droplet size caused their considerable different

acceleration effects on reaction.

In microdroplets, not only the collision frequency increased due to the accumulation effect but also the surface effect, which was proposed to catalyze the microdroplet surface reaction, although experimental investigations are seldom carried out so far as we know. We suggested that H^+ and OH^- could be readily formed at the air-microdroplet interface than in bulk water, due to the lack of three-dimensional hydrogen bonding and the presence of a strong electric field. It is known that a higher H^+ concentration could facilitate acid hydrolysis of proteins and a higher OH^- concentration would promote trypsin activity.

Recently we have confirmed the presence of hydroxyl radicals at the interface by using salicylic acid to capture hydroxyl radicals and also assayed the spontaneous generation of H_2O_2 in microdroplets. Therefore, we added tiny hydrogen peroxide (30 μM) to a myoglobin solution and found the digestion time in bulk could be reduced to 4.5 h from overnight. This test confirmed the formation of reactive oxygen-containing species, such as hydroxyl radicals ($\cdot OH$) and hydrogen peroxide molecules, at the air-water interface may promote protein digestion.

Finally, the proteins are reported to be readily driven to the droplet surface by hydrophobic and electrostatic factors, which may act synergistically with increased collision frequencies and surface effect in microdroplets to improve protein digestion.

Comment 7. The authors claim that the polarity of the voltage has little effect and that even no voltage was effective but to a less digestive extent. I am wondering if a possible digestive mechanism could be that the high ionic state promotes denaturation of the protein which facilitates the reaction acceleration. In figure 2 it is evident that the proteins with the higher ionic states 5, 6 and 7 are cleaved at a

faster rate than the proteins with lower charge state 3 and 4. This seems contradictory.

Response:

This is an interesting question. It is possible that higher charged proteins are digested more efficiently than lower charged states but it would need more investigation. The charge states are basically determined by the original molecule size and their 3D structure in microdroplets. The high voltage is just used for efficient ionization and detection, which is a revelation of the molecular size and structure, instead of a cause. Higher charge states in mass spectra imply that proteins are more unfolded and denatured, and thus easier to digest. This is in line with the data in Figure 2 because ACTH is a big peptide, not a protein. However, we agreed it is possible that high voltage may advance the digestion, but not so much as tested in our case by changing the voltage value.

REVIEWERS' COMMENTS:

Reviewer #1 (Remarks to the Author):

The manuscript "Ultrafast enzymatic digestion of proteins by microdroplet mass spectrometry" appears to be suitable for publication in Nature Communications. The authors answered this reviewers critiques satisfactorily.

Reviewer #2 (Remarks to the Author):

The authors have adequately addressed all of my concerns. However I think the authors should more completely or double check to include the responses into the manuscript itself.